# Hospitalization costs for COVID-19 in Ethiopia: Empirical data and analysis from Addis Ababa's largest dedicated treatment center

**Solomon Tessema Memirie**[1]*, **Amanuel Yigezu**[2], **Samuel Abera Zewdie**[3], **Alemnesh H. Mirkuzie**[2], **Sarah Bolongaita**[4], **Stéphane Verguet**[4]

**1** Addis Center for Ethics and Priority Setting, College of Health Sciences, Addis Ababa University, Addis Ababa, Ethiopia, **2** National Data Management Center for Health, Ethiopian Public Health Institute, Addis Ababa, Ethiopia, **3** Partnership and Cooperation Directorate, Ministry of Health, Addis Ababa, Ethiopia, **4** Department of Global Health and Population, Harvard T.H. Chan School of Public Health, Boston, Massachusetts, United States of America

\* tess_soul@yahoo.com

## Abstract

### Background

The COVID-19 pandemic has caused profound health, economic, and social disruptions globally. We assessed the full costs of hospitalization for COVID-19 disease at Ekka Kotebe COVID-19 treatment center in Addis Ababa, the largest hospital dedicated to COVID-19 patient care in Ethiopia.

### Methods and findings

We retrospectively collected and analysed clinical and cost data on patients admitted to Ekka Kotebe with laboratory-confirmed COVID-19 infections. Cost data included personnel time and salaries, drugs, medical supplies and equipment, facility utilities, and capital costs. Facility medical records were reviewed to assess the average duration of stay by disease severity (either moderate, severe, or critical). The data collected covered the time-period March-November 2020. We then estimated the cost per treated COVID-19 episode, stratified by disease severity, from the perspective of the provider. Over the study period there were 2,543 COVID-19 cases treated at Ekka Kotebe, of which, 235 were critical, 515 were severe, and 1,841 were moderate. The mean patient duration of stay varied from 9.2 days (95% CI: 7.6–10.9; for moderate cases) to 19.2 days (17.9–20.6; for critical cases). The mean cost per treated episode was USD 1,473 (95% CI: 1,197–1,750), but cost varied by disease severity: the mean cost for moderate, severe, and critical cases were USD 1,266 (998–1,534), USD 1,545 (1,413–1,677), and USD 2,637 (1,788–3,486), respectively.

### Conclusions

Clinical management and treatment of COVID-19 patients poses an enormous economic burden to the Ethiopian health system. Such estimates of COVID-19 treatment costs inform financial implications for resource-constrained health systems and reinforce the urgency of

**Data Availability Statement:** Relevant data are within the manuscript and its Supporting Information files. Further relevant data sets are

available on figshare at https://doi.org/10.6084/m9.figshare.17132183.

**Funding:** SV, STM, and SB acknowledge funding support from Gavi, the Vaccine Alliance and the Bill & Melinda Gates Foundation [INV-010174]. The funders of the study had no role in study design, data collection and analysis, decision to publish, or preparation of the manuscript. The views expressed are those of the authors and not necessarily those of the funders. Funder Name: Gavi, the Vaccine Alliance Grant Number: none Grant Recipient: Dr. Stéphane Verguet Funder Name: Bill & Melinda Gates Foundation Grant Number: INV-010174 Grant Recipient: Dr. Stéphane Verguet.

**Competing interests:** The authors declare that they have no competing interests.

implementing effective infection prevention and control policies, including the rapid rollout of COVID-19 vaccines, in low-income countries like Ethiopia.

# Introduction

Coronavirus disease 2019 (COVID-19) has caused widespread health and socioeconomic devastation globally [1,2]. As of May 10, 2021, more than 157 million confirmed COVID-19 cases and 3.2 million deaths were reported globally [1]. Over the same time period, Ethiopia reported more than 262,000 cases and close to 3,900 deaths [1].

The spectrum of COVID-19 disease severity is far-ranging: from asymptomatic to critical illness that requires mechanical ventilation [3]. In most infected individuals, COVID-19 infection causes mild to moderate disease, while approximately 20% develop severe illness with a high risk of mortality [4]. Older age, smoking and comorbidities such as diabetes, hypertension, chronic lung diseases and cancer, have been identified as risk factors for disease severity and death [4]. Severe acute respiratory syndrome, multi-organ dysfunction or complications such as blood-clots are common clinical manifestations in patients with severe or critical COVID-19 infection [5,6]. Similar to other countries, most severe cases and deaths in Ethiopia occurred among patients aged 60 years and above [7].

Since the identification of the first COVID-19 case on March 13, 2020 in Ethiopia, the government has taken multiple infection control and mitigation measures such as school closure, suspension of sports events and public gatherings, alterations of transportation means, community mobilizations and mask mandates, although most of these measures have been loosened over time [7–9]. Despite the pandemic's ongoing burden in Ethiopia, restrictions may have been loosened as a result of the government's inability to sustain its initial stringent mitigation strategies. In low-income countries, such as Ethiopia, where most of the population is employed in the informal sector (only 5.3% of the population participates in wage paying jobs [10]), stay-at-home prevention measures are difficult to apply for prolonged periods [9]. Even though Ethiopia has thus far experienced relatively low numbers of COVID-19 cases, similar to most other African countries, it is among the most affected countries on the continent [1,11,12]. Inadequate COVID-19 testing in Ethiopia likely results in substantial under reporting of cases, obscuring the pandemic's true burden [13].

In order to control COVID-19, the Ethiopian government established quarantine, isolation, COVID-19 testing and treatment facilities in Addis Ababa, the capital city and in all regional states [14]. Furthermore, case management, infection prevention and control protocols were developed and distributed to designated isolation and treatment centers, along with health workforce capacity building measures [14]. Ekka Kotebe treatment center, the largest dedicated COVID-19 treatment center in Ethiopia, was the first facility established for clinical management of Real-Time Polymerase Chain Reaction (RT-PCR) confirmed COVID-19 cases in Ethiopia [15].

The clinical management of patients infected with COVID-19 varies depending on disease severity and pre-existing comorbidities [4,16]. Antiviral treatment using remdesivir is likely to improve recovery and is associated with higher rates of hospital discharge [17]; corticosteroid use is beneficial in reducing short-term mortality and the need for mechanical ventilation [18–20]; and prophylactic antibiotics are indicated in the treatment of severe cases to prevent secondary infections [21]. Patients with critical illness may require mechanical ventilation and admission to intensive care units (ICU), which not only carries substantial costs but has limited availability, as there is a critical shortage of ICU facilities in low-and middle-income

countries like Ethiopia [4,22]. The "second wave" of the pandemic is currently underway in most African countries, including Ethiopia, which could severely overwhelm national health systems [12].

Ethiopia has the second largest population in Africa with an estimated 112 million inhabitants in 2019 [23]. It is a low-income country with a gross domestic product per capita of around USD 900 (2019) [24] and severe financial and infrastructure constraints. Prior to the pandemic, the Ethiopian government spent only USD 33 per capita on health, and there are only 77 physicians per million population (one of the lowest physician densities in the world) [25,26]. As such, even before the COVID-19 pandemic, Ethiopia was facing critical challenges in health service delivery and financing. Therefore, understanding and quantifying the direct economic impact of treating COVID-19 patients is essential for guiding health policy decision making and resource mobilization in Ethiopia and beyond in other sub-Saharan African countries. However, studies on the cost of care of COVID-19 patients are scarce in low- and middle-income countries and in sub-Saharan Africa in particular. In this paper, we aim to estimate the treatment costs associated with inpatient hospital admissions of COVID-19 patients in Ethiopia's largest dedicated treatment center.

## Methods

### Study design and setting

We retrospectively collected clinical and cost data from Ekka Kotebe center, the largest designated COVID-19 treatment facility in Addis Ababa where most of the COVID-19 cases in Ethiopia were treated. The center began accepting COVID-19 patients in mid-March 2020. All patients who were admitted to the center over the period March 15 to November 30 2020 were included in the analysis.

Ekka Kotebe has a total of 310 beds, 12 of which are dedicated to intensive care patients. It possesses an ICU facility that can manage critical cases, and it was the first treatment center established for inpatient clinical care of RT-PCR confirmed COVID-19 cases in Ethiopia. Specific management of cases (during the study period) varied depending on disease severity and was guided by the COVID-19 protocol established by the Ethiopian Public Health Institute (EPHI) [16]. Definitions for cases and case management types are provided in S1 Appendix. Patients were discharged from the center upon complete resolution of symptoms and negative RT-PCR on two consecutive nasopharyngeal and oropharyngeal samples collected 24 hours apart [15].

### Data sources and analysis

Clinical data were collected from patient registries and included the number of patients admitted to the center, the level of disease severity, and the duration of hospital stay. Data on staff salaries, allowances and duty payments, and fuel for transport and utility costs were collected from the finance department of the center. Monthly allowances were provided to all staff working at the center from March to May 2020, which eventually were only paid for physicians due to budget constraints. We used pharmacy data to identify drugs and supplies consumed over the study time period. Similarly, data on laboratory diagnostic tests (e.g. X-ray and ultrasound) were extracted from the respective hopsital departments. Furthermore, the center provided meals for both staff and patients. To account for the capital costs, we quantified building space, number and type of cars (including rental cars), and number and type of medical equipment used at Ekka Kotebe center. All data were collected between March 15 to November 30, 2020 after getting permission from the respective departments at Ekka Kotebe center. Costs were estimated from the healthcare provider perspective and only direct medical costs borne

by the provider were included. To compute costs, we used an ingredients-based approach which calculated costs by multiplying quantities of inputs by their unit prices (all cost calculations are detailed in S2 Appendix). For drugs and supplies, we used price data from the Ethiopian Pharmaceuticals Fund and Supply Agency. At Ekka Kotebe all services were provided free of charge, therefore, costs for laboratory and diagnostic tests were based on data sources from other government health facilities. The cost of meals per serving was obtained from the EPHI, which is the coordinating organization leading the COVID-19 response in Ethiopia [27]. Personnel costs were computed as the sum of salary, allowances, and duty time payments for all the staff in the facility including support staff (administation, housekeeping, and laundry). To estimate the cost of buildings, we measured the total building area in the facility and multiplied it by the local market rental rate estimate, which was based on the average rental price of several buildings in the neighborhood. The cost of vehicles was estimated using their rental equivalents. We calculated the equivalent annual cost (and adjusted for the period) of equipments using initial capital outlay over the lifetime of the asset. Equipment costs were retrieved from the Ekka Kotobe finance department and World Health Organization's (WHO) COVID-19 essential supplies forecasting tool [28]. Data on the useful life of assets, such as medical equipment, were obtained from elsewhere as local data were lacking [29]. We applied a discount rate of 3% per year following WHO guidelines [30]. Total cost was calculated as the sum of costs for personnel, drugs and supplies, laboratory and diagnostic, food and utilities, and capital (e.g., buildings, equipment, vehicles). Costs were converted into 2020 USD using the exchange rate from local currency (Ethiopian Birr) to USD [31].

We computed the average duration of hospital stay (i.e. number of inpatient days) for each level of disease severity (moderate, severe, and critical). We used the number of patients in each severity category and their duration of stay (i.e. total patient days of each category) to apportion the total cost to each category. Subsequently, cost per treated patient and cost per bed day were calculated for each category of disease severity. Furthermore, we computed the average and monthly (March to November, 2020) bed occupancy rate (BOR = total number of inpatient bed-days times 100/available bed-days during the period) to evaluate utilization of the center [32].

## Ethical consideration

Ethical clearance was acquired from the Ethiopian Public Health Institute scientific and ethical review committee (EPHI-IRB-275-2020). We accessed fully anonymized data and patient informed consent were not required.

## Results

A total of 2,543 RT-PCR confirmed COVID-19 cases were admitted and managed in Ekka Kotebe center over the study period (March 15 to November 30, 2020). Most cases were of moderate severity (n = 1,841), followed by severe cases (515) and patients in critical condition (235). Among the 2,543 cases, 131 patients died (i.e. a case fatality ratio of 5.2%) and all the deaths occurred among patients in critical condition and were attributed to COVID-19. Fig 1 summarizes the characteristics of COVID-19 cases, including the number of monthly cases and the average duration of hospital stay, stratified by level of disease severity.

The total cost incurred for COVID-19 treatment at Ekka Kotebe center for the study period was estimated at USD 3.75 million. Fig 2 and Table 1 provide details on monthly costs broken down by cost category, as well as the cost per treated episode for each cost category. Personnel and food costs accounted for nearly 80% of the total costs. Personnel costs were relatively high in the first few months and were largely attributable to variations in staff allowance payments.

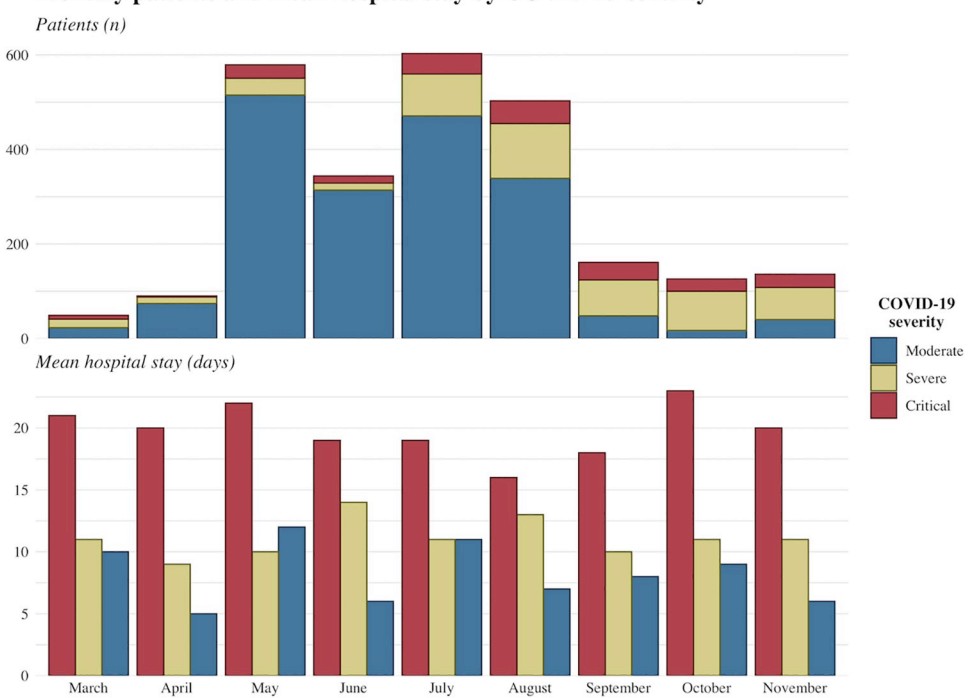

**Fig 1. Number of monthly RT-PCR positive COVID-19 patients admitted to Ekka Kotebe treatment center and mean hospital stay (days) by level of disease severity.**

Monthly allowances were paid to all staff (physicians, health officers, nurses, support staff) working at the center during the initial months of the pandemic, but were eventually only continued for physicians.

The average cost per treated episode and cost per bed day were USD 1,473 (95% CI: 1,197–1,750) and USD 137 (53–221), respectively. The cost per treated episode and cost per bed day varied over the study time period, from USD 968 to USD 7,753 per treated episode and from

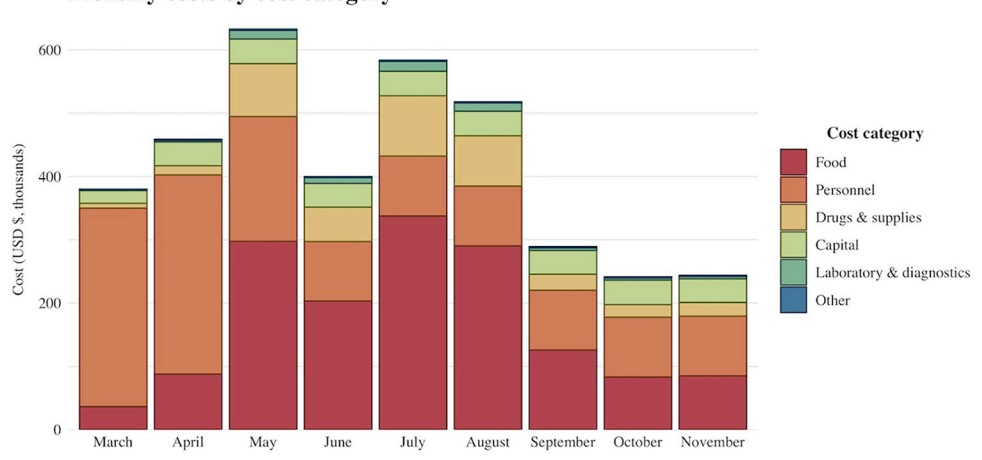

**Fig 2. Total and monthly costs by different cost category in 2020 USD at Addis Ababa Ekka Kotebe COVID-19 treatment center.**

**Table 1. Average cost (in 2020 USD) per treated episode across cost categories.**

| Classification | Mean cost per patient |
|---|---|
| Food | $608 |
| Personnel | $547 |
| Drugs & supplies | $158 |
| Capital | $128 |
| Laboratory & diagnostics | $26 |
| Other | $6 |
| Total | $1,473 |

USD 84 to USD 856 per bed day (Table 2). Furthermore, the cost per bed day varied by level of disease severity: USD 119, USD 145, and USD 247 for moderate, severe and critical (requiring ICU care) patients, respectively. Similarly, we observed large variations in the bed occupancy rate (BOR) ranging from 6 to 75% over the study period. Disaggregation by disease severity found average costs per treated episode of USD 1,266 (95% CI: 998–1,543) for moderate disease, USD 1,545 (1,413–1,677) for severe disease, and USD 2,637 (1,788–3,486) for critical disease requiring ICU admission (Table 3).

Further analysis of costs at different BORs and by level of disease severity show large variations in cost per treated episode and cost per bed day (Table 3). At the lowest observed BOR (6%), the cost per bed day and per treated episode for critical care were USD 1,533 (885–2,180) and USD 8,926 (7,363–10,488), respectively. In contrast, the cost per bed day and cost per treated episode for the highest observed BOR (75%) were much lower at USD 158 (67–249) and USD 2,141 (1,808–2,475), respectively.

## Discussion

This paper reports an analysis of the costs associated with inpatient care for COVID-19 patients at Ekka Kotebe treatment center, which was the first facility established for clinical management of RT-PCR confirmed COVID-19 cases in Ethiopia. Our findings show that the average cost for inpatient care of COVID-19 was USD 1,473 (95% CI: 1,197–1,750) per treated episode over the March-November 2020 study period. Even though evidence on the cost of clinical management of other diseases with high epidemic potential, such as severe acute respiratory syndrome or influenza, is lacking in Ethiopia, the cost of COVID-19 hospitalizaton estimated was substantially higher than those reported for other infectious diseases (e.g. roughly USD 150 per severe measles case and around USD 213 for annual treatment costs for HIV/

**Table 2. Monthly and average inpatient costs per bed day and per treated episode (in 2020 USD), along with bed occupancy rate.**

| | Patients (n) | Inpatient hospital stay (days) | Bed occupancy rate (%) | Total cost | Mean cost per bed-day | Mean cost per patient |
|---|---|---|---|---|---|---|
| March | 49 | 596 | 12 | $379,491 | $637 | $7,753 |
| April | 92 | 536 | 6 | $457,923 | $856 | $4,986 |
| May | 529 | 7,156 | 75 | $628,144 | $88 | $1,196 |
| June | 344 | 2,379 | 26 | $396,851 | $168 | $1,162 |
| July | 603 | 6,977 | 73 | $578,501 | $84 | $968 |
| August | 503 | 4,649 | 48 | $513,671 | $111 | $1,030 |
| September | 161 | 1,810 | 20 | $287,794 | $160 | $1,796 |
| October | 126 | 1,664 | 17 | $240,424 | $145 | $1,917 |
| November | 136 | 1,548 | 17 | $242,685 | $158 | $1,793 |
| Total | 2,543 | 27,315 | 34 | $3,725,484 | $137 (mean) | $1,473 (mean) |

**Table 3. Average cost (in 2020 USD) per treated patient and per bed day at different bed occupancy rates and by level of disease severity.**

| Bed occupancy rate | Moderate | | Severe | | Critical | |
|---|---|---|---|---|---|---|
| | Mean cost per patient | Mean cost per bed-day | Mean cost per patient | Mean cost per bed-day | Mean cost per patient | Mean cost per bed-day |
| 6% | $4,286 | $736 | $5,230 | $898 | $8,926 | $1,533 |
| 34% | $1,266 | $117 | $1,545 | $143 | $2,637 | $244 |
| 75% | $824 | $76 | $1,254 | $93 | $2,141 | $158 |

Moderate = COVID-19 patients with pneumonia.

Severe = COVID-19 patients with severe pneumonia.

Critical = COVID-19 patients with any of the following: acute respiratory distress syndrome, sepsis, or septic shock.

AIDS [33,34]). In our analysis, the mean cost per bed day ranged from USD 119 (for moderate disease) to USD 247 (critical disease requiring ICU care). Comparatively, moderately high costs have been reported in Kenya: from USD 64 per bed day for mild-to-moderate cases (when patients were managed in an isolation center) to USD 600 per bed day for patients admitted to ICU [35].

Our analysis also reported on the costs during various time periods of the pandemic with varying bed occupancy rates (BOR, varying from 6 to 75%). The case management costs were substantially higher earlier in the pandemic when BOR was low. Higher BOR could result in lower patient treatment costs through improved efficiency of health facilities yielding important economies of scale [36,37]. Furthermore, staff allowance payments early in the pandemic (which were discontinued later for most staff at the center) could have contributed to higher costs.

Of note, personnel (37%) and food (41%) costs accounted for nearly 80% of the total costs followed by drug and supply costs (11%). Unlike the findings in our report, a study in China reported drug costs as the major cost drivers, accounting for 45% of overall costs [38]. In this study, clinical management included drugs such as immunomodulators and Chinese medicines that were expensive (these two drugs accounted for 59% of overall drug costs in the study) but such drugs were not part of the treatment protocol in Ethiopia [38]. Besides, the case management protocol in Ethiopia does not recommend the use of newer antivirals, such as remdesivir, but patients with severe illness are given broad spectrum antibiotics (such as azithromycin, ceftriaxone, ceftazidime) for the treatment of superimposed bacterial infection [16].

The generalizability of our study findings may be limited due to the inclusion of a single treatment center, even though Ekka Kotebe was the first and largest hospital designated to manage COVID-19 cases in Ethiopia [7]. There may be variations in the staff mix (most physicians, especially those with specialty and sub-specialty training, reside in Addis Ababa), BOR, and other cost drivers among different COVID-19 centers in Ethiopia that may influence the cost per treated episode. Furthermore, our study did not include COVID-19 treatment costs in private facilities where some COVID-19 patients may seek care. The private health sector is expanding and plays an important role in the Ethiopian health delivery system especially in Addis Ababa where 25 (out of 30) hospitals are private [39]. However, despite their large numbers, public hospitals have substantially higher bed capacities and the vast majority of COVID-19 patients were directed to dedicated treatment centers established by the government, like Ekka Kotebe [15,39]. Additionally, our analysis only included costs from the healthcare provider perspective and therefore misses potentially large out-of-pocket payments for some services that may not have been available at the center, as well as potentially substantial indirect

costs (e.g. time and wage losses) for caregivers and families. We also did not assess cost variations for comorbid conditions due to limited available data on resource use for comorbidities. This is relevant because comorbidies, such as cancer and diabetes, are risk factors for severe COVID-19, and therefore patients who seek COVID-19 related care are more likely to experience additional financial burdens [4,40].

Despite these important shortcomings, our study provides important insight for inpatient care costs of COVID-19 disease by level of severity in Ethiopia. The average cost per treated episode is substantial and imposes considerable financial burden on the already under-resourced health system in Ethiopia, where per capita health expenditure is only USD 33 [25]. Low-income countries, especially those in sub-Saharan Africa, have been severly hit by the economic crisis brought on by the COVID-19 pandemic. This threatens the meaningful progress countries have made toward poverty reduction since 2000 and may widen existing gaps in income inequality, access to education, and gender equality [2]. The economic crisis coupled with the diversion of already meager health resources for the care of patients with COVID-19 could substantially affect the delivery of other essential services such as maternal and child health care and non-communicable diseases management and could compromise aspirations for universal health coverage [41,42]. These findings reiterate the critical importance of rapid vaccine rollout via the COVID-19 Global Vaccine Access Facility (COVAX) [43] for countries in sub-Saharan Africa, not only for curbing the global pandemic and emerging virus variants but to prevent complete collapse of local health systems.

## Supporting information

**S1 Appendix.**
(DOCX)

**S2 Appendix.**
(DOCX)

## Acknowledgments

We are grateful to the Ekka Kotebe treatment center administrative staff and clinicians and the data collection team.

## Author Contributions

**Conceptualization:** Solomon Tessema Memirie, Amanuel Yigezu, Samuel Abera Zewdie.

**Formal analysis:** Solomon Tessema Memirie.

**Funding acquisition:** Alemnesh H. Mirkuzie, Stéphane Verguet.

**Investigation:** Solomon Tessema Memirie.

**Methodology:** Solomon Tessema Memirie, Amanuel Yigezu, Samuel Abera Zewdie.

**Validation:** Solomon Tessema Memirie, Amanuel Yigezu, Samuel Abera Zewdie.

**Visualization:** Solomon Tessema Memirie, Sarah Bolongaita.

**Writing – original draft:** Solomon Tessema Memirie.

**Writing – review & editing:** Solomon Tessema Memirie, Amanuel Yigezu, Samuel Abera Zewdie, Alemnesh H. Mirkuzie, Sarah Bolongaita, Stéphane Verguet.

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
