## [Decision Letter · Decision Letter 0]

23 Aug 2021

PONE-D-21-17483

Hospitalization costs for COVID-19 in Ethiopia: empirical data and analysis from Addis Ababa’s largest dedicated treatment center

PLOS ONE

Dear Dr. Memirie,

Thank you for submitting your manuscript to PLOS ONE. After careful consideration, we feel that it has merit but does not fully meet PLOS ONE’s publication criteria as it currently stands. Therefore, we invite you to submit a revised version of the manuscript that addresses the points raised during the review process.

We look forward to receiving your revised manuscript.

Kind regards,

Sanjay Kumar Singh Patel, Ph.D.

Academic Editor

PLOS ONE

Journal Requirements:

2. In your ethics statement in the Methods section and in the online submission form, please provide additional information about the data used in your retrospective study. Specifically, please ensure that you have discussed whether all data were fully anonymized before you accessed them and/or whether the IRB or ethics committee waived the requirement for informed consent. If patients provided informed written consent to have data from their medical records used in research, please include this information.

4. Thank you for stating the following in the Acknowledgments/ Funding Section of your manuscript: 

We acknowledge funding support from Gavi, the Vaccine Alliance and the Bill & Melinda Gates Foundation (INV-010174). The funder of the study had no role in study design, data collection and analysis, decision to publish, or preparation of the manuscript. The corresponding author had full access to all the data in the study and had final responsibility for the decision to submit for publication.

We acknowledge funding support from Gavi, the Vaccine Alliance and the Bill & Melinda Gates Foundation (INV-010174). The funder of the study had no role in study design, data collection and analysis, decision to publish, or preparation of the manuscript. The corresponding author had full access to all the data in the study and had final responsibility for the decision to submit for publication.

Reviewers' comments:

Reviewer's Responses to Questions

**Comments to the Author**

1. Is the manuscript technically sound, and do the data support the conclusions?

Reviewer #1: Yes

Reviewer #2: Partly

Reviewer #3: Yes

2. Has the statistical analysis been performed appropriately and rigorously? 

Reviewer #1: Yes

Reviewer #2: Yes

Reviewer #3: Yes

3. Have the authors made all data underlying the findings in their manuscript fully available?

Reviewer #1: Yes

Reviewer #2: Yes

Reviewer #3: Yes

4. Is the manuscript presented in an intelligible fashion and written in standard English?

Reviewer #1: Yes

Reviewer #2: Yes

Reviewer #3: Yes

5. Review Comments to the Author

Reviewer #1: This retrospective study by Solomon et al is simple but quite interesting. Being single-centric study the sample size of n=2543 is good for conclusion. However, there are few small things (typos etc.) which can be corrected, if possible:

1. Line no. 50, 55, 150, 183, 185 and 201: Better to use the word "Duration" instead of "Length" of hospital stay.

2. Line no. 94 and 124: Space between "Under-reporting" and "Decision-making".

3. Line no. 103: Real-Time PCR is the gold standard for the SARS-CoV-2 diagnosis, please check whether the positivity was confirmed by Reverse Transcription Polymerase chain Reaction or by One-Step RT-PCR or by RT-PCR.

4.Under the sub-heading Data source and Analysis: We assume that the authors have authority or permission to access the data from the different departments and agencies.

5. Under Result section: line no. 199: Total 131 deaths were reported among the total 2,543 cases. Please clarify how many deaths were reported among the moderate, severe and critical categories individually.

Also, the deaths were due to the underlying conditions or co-morbidity or only due to COVID-19 only.

6. In the Introduction; line no. 75, the authors are suggested to include few lines on the SARS-CoV-2 multiorgan involvement or complications like blood-clots formation which is the contributing factors for the severity or critical situations. Authors can cite the following articles:

1. Thakur, V.; Ratho, R.K.; Kumar, P.; Bhatia, S.K.; Bora, I.; Mohi, G.K.; Saxena, S.K.; Devi, M.; Yadav, D.; Mehariya, S. Multi-Organ Involvement in COVID-19: Beyond Pulmonary Manifestations. J. Clin. Med. 2021, 10, 446. https://doi.org/10.3390/jcm10030446

2. Sourav Biswasa, Vikram Thakur, Parneet Kaur, Azhar Khan, Saurabh Kulshrestha, Pradeep Kumar. Blood clots in COVID-19 patients: Simplifying the curious mystery. https://doi.org/10.1016/j.mehy.2020.110371

3. Sanjay K. S. Patel, Jung-Kul Lee, Vipin C. Kalia. Deploying Biomolecules as Anti-COVID-19 Agents. Indian J Microbiol (July–Sept 2020) 60(3):263–268 https://doi.org/10.1007/s12088-020-00893-4

4. Rishi, P., Thakur, K., Vij, S. et al. Diet, Gut Microbiota and COVID-19. Indian J Microbiol 60, 420–429 (2020). https://doi.org/10.1007/s12088-020-00908-0

Reviewer #2: Summary:

The authors of this paper studied hospitalization costs of for COVID-19 disease at a primary COVID-19 treatment center, Ekka Kotebe, in Addis Ababa, which happens to be the largest hospital dedicated to COVID-19 patient care in Ethiopia. They included personnel time and salaries, drugs, medical supplies and equipment, facility utilities, and capital costs to estimate the treatment cost. They concluded the fact that in a resource-constrained health systems estimates of COVID-19 treatment costs is important and suggested a rapid rollout of COVID-19 vaccines is extremely critical.

Major comments:

1. While this is an important study however it has some drawbacks primarily the study is too narrow. The authors have focused in only one hospital. They mentioned “the private health sector expanding and plays an important role in the Ethiopian health delivery system especially in Addis Ababa where 25 (out of 30) hospitals are private”, however data from those canters were not included. So, based on this fact I believe this study can’t not be generalized.

2. I am surprised that the authors have not include the cost arising duo to comorbid conditions. It has been seen there has been a significant increase in comorbidity due to COVID 19 infections. I would like to know why the authors have considered this cause.

Minor comments:

1. “Prior to the pandemic, the Ethiopian government spent only USD 33 per capita on health, and there are only 77 physicians per million population (one of the lowest physician densities in the world)”. Please provide a reference.

2. The author says, “Costs for laboratory and diagnostic tests were based on data sources from government health facilities.” I am curious to know if there any difference in cost reported by the gov health facility and Ekka Kotebe hospital.

3. Under the method section the author says “ We collected data over the period March 15 to November 30 2020.” Whereas under data source and analysis they says, “All data were collected between August 29 and October 10, 2020”. I am wondering which one is correct?

Reviewer #3: The manuscript by Memirie et al. “Hospitalization costs for COVID-19 in Ethiopia: empirical data and analysis from Addis Ababa’s largest dedicated treatment center” is very interesting. It can be accepted for the publication in PLOS ONE. The citations should be in standard format i.e., Journal name abbreviations. The quality of Tables and Figures should be improved.

---

## [Author Response · Author response to Decision Letter 0]

17 Oct 2021

September 16, 2021

Dr. Sanjay Kumar Singh Patel

Academic Editor

PLOS ONE

Dear Editors,

We are submitting our revised manuscript entitled "Hospitalization costs for COVID-19 in Ethiopia: empirical data and analysis from Addis Ababa’s largest dedicated treatment center" (PONE-D-21-17483) to PLOS ONE. 

We would like to thank the editor and reviewers for the constructive comments and suggestions which have helped to strengthen our work substantially. 

We have edited the manuscript to conform to PLOS ONE’s style requirments and checked our reference list for completeness and retracted articles (we did not find any retracted articles). We have also provided additional information in our ethics statement. Currently, our acknowledgement section does not include funding and conflict of interest statements.

Additionally, we would like to include the following funding statement in the manuscript: “SV, STM, and SB acknowledge funding support from Gavi, the Vaccine Alliance. The funder had no role in study design, data collection and analysis, decision to publish, or preparation of the manuscript. The views expressed are those of the authors and not necessarily those of the funder.”

Below, we have provided a detailed point-by-point response to all comments and suggestions raised in the review.

Thank you very much for your consideration of our manuscript.

Sincerely,

Solomon Tessema Memirie, on behalf of the authors

 

Point-by-point response to the reviewers’ comments

Reviewer #1:

This retrospective study by Solomon et al is simple but quite interesting. Being single-centric study the sample size of n=2543 is good for conclusion. However, there are few small things (typos etc.) which can be corrected, if possible:

Authors’ reponse: We thank the reviewer for his/her positive comments on our manuscript and the additional comments are addressed below.

1. Line no. 50, 55, 150, 183, 185 and 201: Better to use the word "Duration" instead of "Length" of hospital stay.

Authors’ response: We have now used “Duration” instead of “Length”. Please see line numbers: 50, 55, 153, 186, 188, 207.

2. Line no. 94 and 124: Space between "Under-reporting" and "Decision-making".

Authors’ response: We have now included a space. Please see line numbers: 95 and 125.

3. Line no. 103: Real-Time PCR is the gold standard for the SARS-CoV-2 diagnosis, please check whether the positivity was confirmed by Reverse Transcription Polymerase chain Reaction or by One-Step RT-PCR or by RT-PCR.

Authors’ response: Thank you for raising this important point. We have now checked and the COVID-19 test was Real-Time PCR. We have now modified the sentence as follows: “Ekka Kotebe treatment center, the largest dedicated COVID-19 treatment center in Ethiopia, was the first facility established for clinical management of Real-Time Polymerase Chain Reaction (RT-PCR) confirmed COVID-19 cases in Ethiopia [13].” Please see line numbers: 105-106.

4. Under the sub-heading Data source and Analysis: We assume that the authors have authority or permission to access the data from the different departments and agencies.

Authors’ response: Thank you for raising this point. We have modified the last statement in Data source and Analysis as follows: “All data were collected between March 15 to November 30, 2020 after getting permission from the respective departments at Ekka Kotebe center.” Please see line numbers 162-163.

5. Under Result section: line no. 199: Total 131 deaths were reported among the total 2,543 cases. Please clarify how many deaths were reported among the moderate, severe and critical categories individually. Also, the deaths were due to the underlying conditions or co-morbidity or only due to COVID-19 only.

Authors’ response: Thank you for this important point. We have now modified the statement as follows: “Among the 2,543 cases, 131 patients died (i.e. a case fatality ratio of 5.2%) and all the deaths occurred among patients in critical condition and were attributed to COVID-19.” Please see line numbers 204-205.

6. In the Introduction; line no. 75, the authors are suggested to include few lines on the SARS-CoV-2 multiorgan involvement or complications like blood-clots formation which is the contributing factors for the severity or critical situations. Authors can cite the following articles:

1. Thakur, V.; Ratho, R.K.; Kumar, P.; Bhatia, S.K.; Bora, I.; Mohi, G.K.; Saxena, S.K.; Devi, M.; Yadav, D.; Mehariya, S. Multi-Organ Involvement in COVID-19: Beyond Pulmonary Manifestations. J. Clin. Med. 2021, 10, 446. https://doi.org/10.3390/jcm10030446

2. Sourav Biswasa, Vikram Thakur, Parneet Kaur, Azhar Khan, Saurabh Kulshrestha, Pradeep Kumar. Blood clots in COVID-19 patients: Simplifying the curious mystery. https://doi.org/10.1016/j.mehy.2020.110371

3. Sanjay K. S. Patel, Jung-Kul Lee, Vipin C. Kalia. Deploying Biomolecules as Anti-COVID-19 Agents. Indian J Microbiol (July–Sept 2020) 60(3):263–268 https://doi.org/10.1007/s12088-020-00893-4

4. Rishi, P., Thakur, K., Vij, S. et al. Diet, Gut Microbiota and COVID-19. Indian J Microbiol 60, 420–429 (2020). https://doi.org/10.1007/s12088-020-00908-0

Authors’ response: We have now added the following sentence in the Introduction: "Severe acute respiratory syndrome, multi-organ dysfunction or complications such as blood-clots are common clinical manifestations in patients with severe or critical COVID-19 infection [5,6].". We have also cited two of the suggested articles. Please see line numbers 80-82 and references 5 and 6.

Reviewer #2

The authors of this paper studied hospitalization costs of for COVID-19 disease at a primary COVID-19 treatment center, Ekka Kotebe, in Addis Ababa, which happens to be the largest hospital dedicated to COVID-19 patient care in Ethiopia. They included personnel time and salaries, drugs, medical supplies and equipment, facility utilities, and capital costs to estimate the treatment cost. They concluded the fact that in a resource-constrained health systems estimates of COVID-19 treatment costs is important and suggested a rapid rollout of COVID-19 vaccines is extremely critical.

Authors’ response: We thank the reviewer for his/her constructive comments on our manuscript and the additional comments are addressed below.

Major comments:

1. While this is an important study however it has some drawbacks primarily the study is too narrow. The authors have focused in only one hospital. They mentioned “the private health sector expanding and plays an important role in the Ethiopian health delivery system especially in Addis Ababa where 25 (out of 30) hospitals are private”, however data from those canters were not included. So, based on this fact I believe this study can’t not be generalized.

Authors’ response: Thank you for raising this important point. The predominant majority of COVID-19 patients were managed in public facilities in Ethiopia including in Addis Ababa. Even though the number of private hospitals was higher as compared to public hospitals, the public facilities have substantially higher bed numbers. Furthermore, unlike private facilities, the government established several dedicated COVID-19 centers for both isolation and clinical management of cases to tackle the pandemic in the country. Regardless, we have acknowledged that not including private facilities is one of the limitations in our study. Please see line numbers 289-290. 

2. I am surprised that the authors have not include the cost arising duo to comorbid conditions. It has been seen there has been a significant increase in comorbidity due to COVID 19 infections. I would like to know why the authors have considered this cause.

Authors’ response: We thank the reviewer for this important comment. We acknowledge that this is one of the limitations of our study. We have now modified the statement as follows to clarify the concern raised by the reviewer: “We also did not assess cost variations for comorbid conditions due to limited available data on resource use for comorbidities. This is relevant because comorbidies, such as cancer and diabetes, are risk factors for severe COVID-19, and therefore patients who seek COVID-19 related care are more likely to experience additional financial burdens [4,40].” Please see lines 297-299.

Minor comments:

1. “Prior to the pandemic, the Ethiopian government spent only USD 33 per capita on health, and there are only 77 physicians per million population (one of the lowest physician densities in the world)”. Please provide a reference.

Authors’ response: We have now included references (we moved “References 25 and 26” up to their appropriate place from the sentence below). Please see line number 123.

2. The author says, “Costs for laboratory and diagnostic tests were based on data sources from government health facilities.” I am curious to know if there any difference in cost reported by the gov health facility and Ekka Kotebe hospital.

Authors’ response: At Ekka Kotebe COVID-19 treatment center all services were provided by the government free of charge. Therefore, we used estimates of laboratory and diagnostic costs from other government facilities. We have now modified the sentence as follows to clarify the point: “At Ekka Kotebe all services were provided free of charge, therefore, costs for laboratory and diagnostic tests were based on data sources from other government health facilities.” Please see line numbers 168-170.

3. Under the method section the author says “ We collected data over the period March 15 to November 30 2020.” Whereas under data source and analysis they says, “All data were collected between August 29 and October 10, 2020”. I am wondering which one is correct?

Authors’ response: We thank the reviewer for pointing out these confusing statements. We have now modified the statement as follows: “All patients who were admitted to the center over the period March 15 to November 30 2020 were included in the analysis.“ Please see lines 138-139.

Reviewer #3

The manuscript by Memirie et al. “Hospitalization costs for COVID-19 in Ethiopia: empirical data and analysis from Addis Ababa’s largest dedicated treatment center” is very interesting. It can be accepted for the publication in PLOS ONE. The citations should be in standard format i.e., Journal name abbreviations. The quality of Tables and Figures should be improved.

Authors’ response: We thank the reviewer for their positive comments on our manuscript. We have now tried to stansardize journal name abbreviations and improve the quality of tables and figures. Please see “References”, “Figure 1”, “Figure 2” and Tables in the manuscript.

---

## [Decision Letter · Decision Letter 1]

22 Nov 2021

Hospitalization costs for COVID-19 in Ethiopia: empirical data and analysis from Addis Ababa’s largest dedicated treatment center

PONE-D-21-17483R1

Dear Dr. Memirie,

We’re pleased to inform you that your manuscript has been judged scientifically suitable for publication and will be formally accepted for publication once it meets all outstanding technical requirements.

Kind regards,

Sanjay Kumar Singh Patel, Ph.D.

Academic Editor

PLOS ONE

---

## [Editor Report · Acceptance letter]

15 Dec 2021

PONE-D-21-17483R1 

Hospitalization costs for COVID-19 in Ethiopia: empirical data and analysis from Addis Ababa’s largest dedicated treatment center 

Dear Dr. Memirie:

I'm pleased to inform you that your manuscript has been deemed suitable for publication in PLOS ONE. Congratulations! Your manuscript is now with our production department. 

Kind regards, 

on behalf of

Dr. Sanjay Kumar Singh Patel 

Academic Editor

PLOS ONE